

# The value of lymphocyte to monocyte ratio in the prognosis of head and neck squamous cell carcinoma: a meta-analysis

Deyou Wei[1,*], Jiajia Liu[1,*] and Jipeng Ma[2]

[1] Department of Otolaryngology, Yantai Hospital of Traditional Chinese Medicine, Yantai, China
[2] Department of Oncology, Yantai Hospital of Traditional Chinese Medicine, Yantai, China
* These authors contributed equally to this work.

Corresponding author
Jipeng Ma, muwoji19@163.com

## ABSTRACT

**Objectives:** Although lymphocyte-monocyte ratio (LMR) is a potential prognostic biomarker in many tumor indications, a doubt occurs around its association with head and neck squamous cell carcinoma (HNSCC). We aimed to evaluate the predictive value of LMR in patients with HNSCC.

**Methods:** We searched PubMed, Web of Science, EMBASE, and the Cochrane database from inception to May 8, 2023 for systematic review and meta-analysis on LMR and outcomes related to HNSCC development. STATA software was used to estimate the correlation between LMR and prognosis. The risk ratio (hazard ratio, HR) and 95% confidence interval l (CI) for overall survival (OS), disease-free survival (DFS), cancer-specific survival (CSS), and progression-free survival (PFS) were calculated, and the association between LMR and OS was further validated by subgroup analysis. The source of heterogeneity with the results of subgroup analysis was analyzed by meta-regression analysis. This meta-analysis was registered at PROSPERO (CRD42023418766).

**Results:** After a comprehensive exploration, the results of 16 selected articles containing 5,234 subjects were evaluated. A raised LMR was connected to improved OS (HR = 1.36% CI [1.14–1.62] $P$ = 0.018), DFS (HR = 0.942, 95% CI [0.631–1.382], $P$ = 0.02), and PFS (HR = 0.932, 95% CI [0.527–1.589], $P$ < 0.022). Subgroup analysis indicated that patients with a low LMR level had a poor prognosis with a critical value of ≥4. The LMR was found to be prognostic for cases with an LMR of <4. The meta-regression analysis showed that the cut-off values and treatment methods were the primary sources of high heterogeneity in patients with HNSCC.

**Conclusions:** Our study suggested that an elevated LMR is a potential prognostic biomarker in patients with HNSCC and could be used to predict patient outcomes.

## INTRODUCTION

Head and neck squamous cell carcinoma (HNSCC) is squamous cell carcinoma in the oral and maxillofacial region, the throat, and the upper respiratory and digestive tract (*Psyrri et al., 2021*). HNSCC is the sixth most common malignant tumor, with a worldwide

incidence of 3.9% (*Kałafut et al., 2021*). Previous studies have indicated that smoking, drinking, poor oral hygiene, and genetic factors are important risk factors for head and neck tumors (*Kordbacheh & Farah, 2021*). In 2018, there were 890,000 new cases of HNSCC and 450,000 deaths from the disease worldwide (*Peyrade et al., 2021*). HNSCC is one of the most common tumors in China (*Song et al., 2021*). Due to the challenging locations of HNSCC, treatment can be complex, and lymph node metastasis is common, contributing to a poor level of 5-year survival (*Zhong, Zhou & Zhu, 2021*). Distant metastasis is a poor prognostic indicator in HNSCC, and the lungs, liver, and bones (spine, skull, ribs, long bones) are the most common sites of metastasis in HNSCC (*Murphy et al., 2022*). Treatment options are usually palliative once patients have progressed to metastatic disease (*Aarstad et al., 2020*). Therefore, it is urgent to acquire improved predictive biomarkers for clinical therapy.

The treatment of HNSCC includes surgery, radiotherapy, chemotherapy, or combination therapy (*Ghosh, Shah & Johnson, 2022*). There is a need to develop new treatments that are highly effective and well tolerated by patients (*Wang et al., 2020*). The tumor-node-metastasis (TNM) staging system is the golden standard for cancer staging but is limited in identifying tumor heterogeneity (*He et al., 2021*). Biomarkers play critical roles in the prognosis of patients with HNSCC and could significantly enhance the prognostic accuracy of the TNM system (*Dankbaar & Pameijer, 2021*). There is currently an unmet need for independent staging criteria in HNSCC to facilitate the disease's diagnosis, prognosis, and treatment (*Ausoni et al., 2016*). Inflammatory cells and proteins have crucial regulatory functions in cancer onset and progression and could also be valuable as prognostic biomarkers (*Rubinstein-Achiasaf et al., 2021*). Recent studies have shown that the ratio of peripheral blood lymphocytes to monocytes (LMR) has prognostic value in various cancers (*Lin et al., 2022*; *Meng et al., 2022*). The association between different hematological markers, such as the ratio of neutrophils to lymphocytes (NLR) and the rate of lymphocytes to monocytes (LMR) is known to have predictive value in various cancers. Hematological indicators are cheaper, easy to analyze, and can be routinely used in a clinical setting (*Nechita et al., 2021*). Recently, it has been reported that LMR plays a crucial role in the prognosis of head and neck malignant tumors (*Yamamoto, Kawada & Obama, 2021*; *Nøst et al., 2021*; *Wu et al., 2017*). However, at present, the prognosis of LMR in patients with HNSCC is not clear. The prognostic value of LMR in HNSCC must be clarified with inconclusive data from several reports. The purpose of this study is to evaluate the relationship between LMR and the prognosis of HNSCC patients.

In this study, we conducted a meta-analysis to assess the prognostic value of the lymphocyte-to-monocyte ratio in patients with HNSCC. We systematically reviewed the literature to assess LMR's diagnostic and prognostic efficacy in patients with HNSCC. The PRISMA statement was followed when conducting and recording the procedure assessment.

## MATERIALS AND METHODS

### Searching scheme

The analysis undertaken in this study was performed according to the Preferred Reporting Items for Systematic Reviews and Meta-Analyses (PRISMA) statement (*O'Dea et al., 2021*). The search terms used in PubMed, the Web of Science, EMBASE, and the Cochrane Library databases from inception to May 8, 2023, were "Squamous Cell Carcinoma of Head and Neck", "Oral Squamous Cell Carcinoma", "Head and Neck Squamous Cell Carcinoma" "Oral Tongue Squamous Cell Carcinoma", "Laryngeal Squamous Cell Carcinoma", "Squamous Cell Carcinoma of the Larynx", "Hypopharyngeal Squamous Cell Carcinoma", "LMR", "lymphocyte to monocyte ratio", "lymphocyte monocyte ratio", "prognosis", "outcome", "survival", "mortality", "laryngeal squamous cell carcinoma", "squamous cell carcinoma of the larynx", "hypopharyngeal squamous cell carcinoma", and "oropharyngeal squamous cell carcinoma". The specific search strategy was (Squamous Cell Carcinoma of Head and Neck or Oral Squamous Cell Carcinoma or Head and Neck Squamous Cell Carcinoma) AND (LMR or lymphocyte to monocyte ratio or lymphocyte monocyte ratio) AND (prognosis or outcome or survival or mortality). The Meta-analysis was registered at PROSPERO (CRD42023418766).

### Eligibility criteria

The inclusion criteria were: (1) Randomized controlled trial or observational studies; (2) HNSCC confirmed by pathological detection; (3) studies that evaluated the prognostic value of LMR in HNSCC; (4) studies that reported the total survival rate (OS), and disease-free survival rate (DFS) or cancer-specific survival (CSS) or other outcome indexes, and provided a hazard ratio (HR), 95% confident interval (CI) or other data such as survival curves.

The exclusion criteria were: (1) Review articles, case reports, letters, or meeting summaries; (2) studies not related to LMR or HNSCC; (3) Repeated publications or studies with similar data; (4) studies with insufficient data to calculate the HR for OS and the corresponding 95% CI; (5) studies were lack of original data of outcome measures in the eligibility criteria. Two researchers (DYW and JPM) independently reviewed the titles and abstracts of the articles to identify the relevant studies.

### Literature search and data extraction

Two investigators (DYW and JJL) independently obtained data according to the inclusion and exclusion criteria. Differences in the identification of studies were resolved through discussion. The extracted data included the first author's name, time, sample size, patient gender, staging method, treatments, median follow-up time, LMR value, outcome index, HR, and 95% CI.

### Literature value assessment

According to the evaluation criteria of the Newcastle-Ottawa quality scale (NOS) (*Lo, Mertz & Loeb, 2014*), studies with total scores of ≥6 were regarded as high quality. The quality of included studies was also independently assessed by two evaluators.
## Statistical analysis

The included studies were analyzed using STATA 12.0 (*Zhang et al., 2020*) and RevMan5.3 (*Hu et al., 2020*) (Cochrane Collaboration) software. The association between LMR and prognostic value in HNSCC was evaluated by HR and the 95% CI. The Cochrane Q test and $I^2$ statistical tests were used to evaluate heterogeneity. Significant difference was defined as $P < 0.10$ (Q-statistic) or $I^2 > 50\%$ and a random effect model was used. If $P > 0.1$ or $I^2 < 50\%$, it suggested that there existed no heterogeneity, and we used the fixed-effects model to combine the data. When $I^2 \geq 50\%$ and $P \leq 0.05$, the heterogeneity was significant; then, we conducted the meta-regression analysis to observe the source of heterogeneity. Subgroup analysis was conducted to evaluate the source of heterogeneity or to prove the meta-analysis's findings further. A sensitivity examination was used to evaluate the stability of the results. A Begg funnel chart and Egger's test were applied to identify publication bias. *P*-values of < 0.05 were set to indicate significant publication bias.

# RESULTS

## Search consequences

PubMed, Web of Science, EMBASE, and the Cochrane Library databases were searched online from inception to May 8, 2023. A total of 157 articles were identified, 45 repeated articles were excluded, and 60 were rejected after reviewing the titles and abstracts. Thirty-six other article were excluded as no result indices were provided. A total of 16 studies were included in our meta-analysis that, included data from 5,234 patients.
The details of the included studies are summarized in Table 1. A total of eight of the studies was from China, and 1 study was included from Japan, India, Thailand, the USA, Korea, and Sudan. All the studies were retrospective analyses in which 16 of the studies described OS, five described PFS, two described CSS, and seven reported DFS. For several articles, distinctive effect sizes were described in their stratified analysis and the effect sizes for these articles were identified. The workflow used in this study is presented in Fig. 1.

## Quality of included studies

The data summarized in Table 2 showed a high inter-rater agreement for the risk of bias assessment (κ between 0.645 and 1.00 across domains). Except for 1 study, all the studies were open-label, and the primary outcome was overall survival. Furthermore, we hypothesized that all studies were at a high risk of bias regarding incomplete outcome data and selective outcome reporting.

## Connection between LMR and OS in HNSCC

A forest map was constructed to explore the relationship between LMR and OS rate in HNSCC patients. The meta-analysis integrated data from 16 studies for LMR and OS. Due to the high heterogeneity amongst the included studies, the data were analyzed using the random effect model. The results showed that the patients with a higher LMR ratio had improved OS (HR =1.36, 95% CI [1.41–1.65] $P = 0.018$) with high heterogeneity ($I^2 = 59.6\%$). The forest map and a summary of the LMR endpoint statistics (OS) are

**Table 1 Summary of the main characteristics of the studies included in the analysis.**

| Author (year) | Country | Study design | Cases | Female/male | Cut-off | Survival outcome | Nos | Follow-up (months)/ median (range) | HR |
|---|---|---|---|---|---|---|---|---|---|
| Jun Aoyama (*Hu et al., 2020*) | Japan | Retrospective | 100 | 11/89 | 1.994 | PFS/OS | 8 | 9 (1–65) | U/M |
| Tarun Jindal (*Mulder et al., 2021*) | India | Retrospective | 126 | – | 3 | CSS | 8 | 18 (2–74) | U/M |
| Jiechao Yang (*Chen et al., 2022*) | China | Retrospective | 197 | 2/195 | 2.98 | OS/CSS/DFS | 9 | 30.95 (1–82) | U/M |
| Pasawat (*Cohen et al., 2019*) | Thailand | Retrospective | 211 | 72/139 | 4 | OS | 8 | – | U/M |
| Qian Song (*Xia et al., 2021*) | China | Retrospective | 680 | 98/582 | 3.17 | DFS/OS | 7 | 61 (56–67) | U/M |
| Chuang (*Zhu et al., 2022*) | China | Retrospective | 141 | 8/133 | 2.99 | PFS/OS | 7 | 45.8 (3–91) | U/M |
| Meng Ding (*Fan et al., 2022*) | China | Retrospective | 493 | 223/261 | 3.4 | OS/DFS | 7 | – | U/M |
| Yi-Wei Lin (*Slagter et al., 2022*) | China | Retrospective | 169 | 76/93 | 4.15 | OS | 7 | – | U/M |
| Kosei Kubota (*Fairfield et al., 2021*) | USA | Retrospective | 183 | 100/305 | 5 | OS/DFS | 6 | 66 (26–93) | U/M |
| Xiang Wu (*D'Orso, Hyder & Mccann, 2020*) | China | Retrospective | 486 | 260/226 | 4 | OS/DFS | 6 | – | U/M |
| Huijun Chen (*Duan et al., 2022*) | China | Retrospective | 473 | 17/456 | 5 | OS/PFS | 6 | – | U/M |
| Hyeon Koh (*Caziuc et al., 2020*) | Korea | Retrospective | 68 | 4/64 | 2.51 | OS | 6 | 66 (47–84) | U/M |
| Youfang Xun (*Ivankova et al., 2021*) | China | Retrospective | 151 | 4/147 | 0.18 | PFS/OS | 7 | 65 (44–84) | U/M |
| Eltohami (*Xu et al., 2021*) | Sudan | Retrospective | 613 | 57/556 | 4.85 | OS/DFS | 7 | – | U/M |
| Paolo Boscolo-Rizzo (*Sznurkowski et al., 2020*) | Italy | Retrospective | 925 | 246/679 | | OS/DFS | | 53 (31–82) | U/M |
| Michael Pogorzelski (*Sumbayev et al., 2020*) | Germany | Retrospective | 218 | 45/173 | | PFS/OS | | – | U/M |

presented in Fig. 2 and Table 3. A Begg's funnel map for the HR of OS showed no indication of publication bias (Fig. 3, *P* < 0.05).

## Connection between LMR and DFS in HNSCC

Data from six studies were integrated into the meta-analysis for LMR and DFS in HNSCC. Due to the high heterogeneity amongst the included studies, the random effect model was used for the analysis. The LMR was strongly associated with improved DFS in HNSCC with0.942 (HR = 0.942, 95% CI [0.631–1.382], *P* = 0.02), with high heterogeneity ($I^2$ = 56.8%). The forest map and the summary of LMR endpoint statistics (DFS) are shown in Fig. 4 and Table 4. Publication bias was evaluated by observing the extent of funnel diagram asymmetry. Subsequently, we employed Begg's test to validate the graphical view from the funnel diagram. *P* < 0.1 was regarded as an indication of publication bias. We observed no publication bias in DFS (*P* = 0.385, Fig. 5). When there was evidence of publication bias, we adjusted the effect sizes using the trim-and-fill method. When there existed publication bias, we used the trim-and-fill method to modify the effect. A sensitivity analysis was conducted to investigate the stability of the data. None of the studies validated a significant effect on the pooled value, indicating that the studies had good stability.
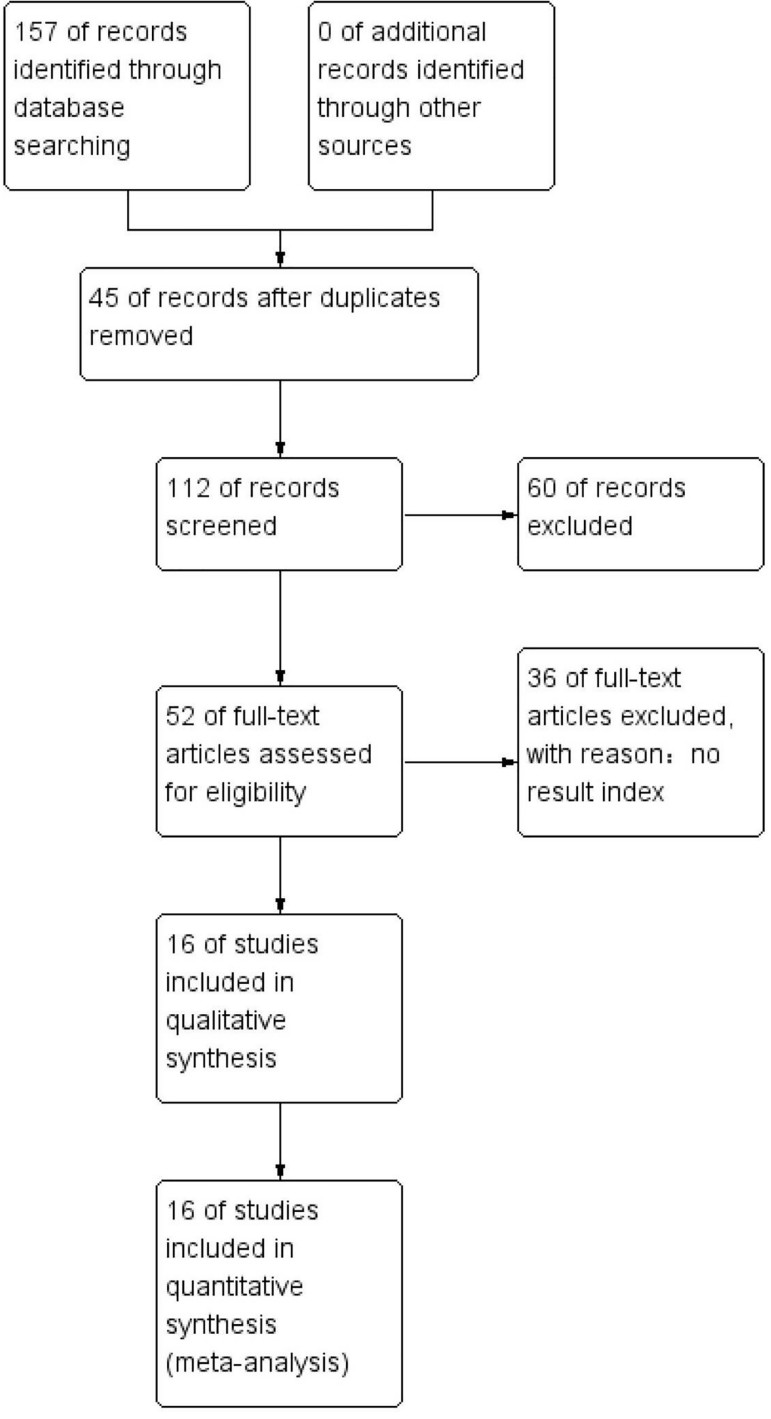

**Figure 1** **Overview of the literature screening process used in this study and a summary of the main results.**

## Connection between LMR and PFS in HNSCC

Data from six studies were integrated into the meta-analysis for LMR and PFS in HNSCC. Due to the high heterogeneity amongst the included studies, the random effect model was

**Table 2 Summary of the risk of bias in the randomized controlled trials identified in this study.**

| Study | Year | Randomization | Allocation concealment | Blinding of participants and staff | Blinding of outcome assessors | Incomplete outcome data | Selective outcome reporting | Other sources of bias |
|---|---|---|---|---|---|---|---|---|
| Jun Aoyama | 2021 | Low | Low | High | High | High | High | Low |
| Tarun Jindal | 2021 | Low | Low | High | High | High | High | Low |
| Jiechao Yang | 2018 | Low | Unclear | High | High | High | High | Low |
| Pasawat | 2022 | Low | Low | Low | Low | High | High | Low |
| Qian Song | 2019 | Low | Low | High | High | High | High | Low |
| Chuang | 2020 | Low | Low | High | High | High | High | Low |
| Meng Ding | 2021 | Low | Low | High | High | High | High | Low |
| Yi-Wei Lin | 2021 | Low | Low | High | High | High | High | Low |
| Kosei Kubota | 2022 | Low | Low | High | High | High | High | Low |
| Xiang Wu | 2021 | Low | Low | High | High | High | High | Low |
| Huijun Chen | 2020 | Low | Unclear | High | High | High | High | Low |
| HYEON KOH | 2021 | Low | Low | Low | Low | High | High | Low |
| Youfang Xun | 2019 | Low | Low | High | High | High | High | Low |
| Eltohami | 2018 | Low | Low | High | High | High | High | Low |
| Paolo Boscolo-Rizzo | 2022 | Low | Low | High | High | High | High | Low |
| Michael Pogorzelski | 2021 | Low | Low | High | High | High | High | Low |
| *Kappa* | NA | 1 | 1 | 0.645 | 0.645 | 1 | 1 | 1 |

Note:
NA, not applicable.

used for further analysis. The LMR was strongly associated with improved DFS in HNSCC with an HR of 0.932 (HR = 0.932, 95% CI [0.527–1.589], $P$ = 0.022), with high heterogeneity ($I^2$ = 64.8%). The forest map and the summary of LMR endpoint statistics (PFS) are shown in Fig. 6 and Table 5. A Begg's funnel map for HR of PFS showed no indication of publication bias (Fig. 7, $P$ < 0.05). As shown in Fig. 7, we observed no publication bias in PFS ($P$ = 0.264), indicating no substantial funnel plot asymmetry.

## Subgroup analysis and meta-regression

The sources of heterogeneity in the survival of patients with HNSCC were explored by subgroup analysis and meta-regression analysis. The heterogeneity of survival studies was relatively high in PFS ($I^2$ = 64.8%) and DFS ($I^2$ = 56.8%), and OS ($I^2$ = 59.6%). After finding the primary sources of heterogeneity in different indicators, a subgroup analysis was performed, and the data are summarized in Table 6. The subgroup analysis showed that the patients with a low LMR level had a poor prognosis with a critical LMR of ≥4.

The meta-regression results showed that the cut-off values and treatment methods were the primary sources of high heterogeneity in patients with HNSCC (Table 7).

## DISCUSSION

According to the World Health Organization (WHO) report in 2019, cancer has become the leading cause of death worldwide (*Wang, 2009*). Head and neck cancer is one of the

| STUDY ID | Year | Cutoff | Harzard Ration<br>M-H,Fixed,95% CI | OS<br>HR(95% CI) | Weight(%) |
|---|---|---|---|---|---|
| JUN AOYAMA | 2021 | 1.994 | | 0.389(0.234-0.632) | 6.34 |
| Tarun Jindal | 2021 | 3 | | 0.742(0.623-1.041) | 7.11 |
| Jiechao Yang | 2018 | 2.98 | | 0.55(0.37-0.81) | 8.09 |
| Pasawat | 2022 | 4 | | 1.51 (1.07,2.12) | 5.06 |
| Qian Song | 2019 | 3.17 | | 0.895(0.816-0.981) | 5.63 |
| Chuang | 2020 | 2.99 | | 1.941 (1.223-3.081) | 9.84 |
| Meng Ding | 2021 | 3.4 | | 0.414 (0.263-0.650) | 5.16 |
| Yi-Wei Lin | 2021 | 4.15 | | 0.408(0.177-0.940) | 7.54 |
| Kosei Kubota | 2022 | 5 | | 0.99 (0.48–2.06) | 5.80 |
| Xiang Wu | 2021 | 4 | | 0.57(0.080-1.097) | 3.39 |
| Huijun Chen | 2020 | 5 | | 1.302 (0.75-2.69) | 9.82 |
| HYEON KOH | 2021 | 2.51 | | 0.35(0.090-1.037) | 3.88 |
| Youfang Xun | 2019 | 0.18 | | 0.07(0.02-0.19) | 8.65 |
| Eltohami | 2018 | 4.85 | | 1.463 (1.102-1.942) | 4.63 |
| Paolo Boscolo-Rizzo | 2022 | 2.92 | | 1.56(1.24–1.95) | 4.57 |
| Michael Pogorzelski | 2021 | 1.55 | | 1.41 (1.05-1.92) | 4.50 |
| Overall ($I^2$= 59.6%, P=0.018) | | | | 1.36 (1.14-1.62) | 100 |

0    1    2    3    4

**Figure 2 A forest map showing the relationship between the LMR and the overall survival of patients with HNSCC in the 13 identified studies.**

**Table 3 A summary of the LMR endpoint statistics (OS).**

| Study ID | Year | Cutoff | OS<br>HR [95% CI] | Weight<br>(%) | P-value |
|---|---|---|---|---|---|
| Jun Aoyama | 2021 | 1.994 | 0.389 [0.234–0.632] | 6.34 | <0.001 |
| Tarun Jindal | 2021 | 3 | 0.742 [0.623–1.041] | 7.11 | 0.04 |
| Jiechao Yang | 2018 | 2.98 | 0.55 [0.37–0.81] | 8.09 | 0.002 |
| Pasawat | 2022 | 4 | 1.51 [1.07,2.12] | 5.06 | 0.019 |
| Qian Song | 2019 | 3.17 | 0.895 [0.816–0.981] | 5.63 | 0.018 |
| Chuang | 2020 | 2.99 | 1.941 [1.223–3.081] | 9.84 | 0.005 |
| Meng Ding | 2021 | 3.4 | 0.414 [0.263–0.650] | 5.16 | <0.001 |
| Yi-Wei Lin | 2021 | 4.15 | 0.408 [0.177–0.940] | 7.54 | 0.035 |
| Kosei Kubota | 2022 | 5 | 0.99 [0.48–2.06] | 5.80 | 0.98 |
| Xiang Wu | 2021 | 4 | 0.57 [0.080–1.097] | 3.39 | <0.001 |
| Huijun Chen | 2020 | 5 | 1.302 [0.75–2.69] | 9.82 | <0.001 |
| Hyeon Koh | 2021 | 2.51 | 0.35 [0.090–1.037] | 3.88 | 0.057 |
| Youfang Xun | 2019 | 0.18 | 0.07 [0.02–0.19] | 8.65 | <0.001 |
| Eltohami | 2018 | 4.85 | 1.463 [1.102–1.942] | 4.63 | 0.0086 |
| Paolo Boscolo-Rizzo | 2022 | 2.92 | 1.56 [1.24–1.95] | 4.57 | |

**Note:**
HR, hazard ratio; OS, overall survival; CI, confidence interval.

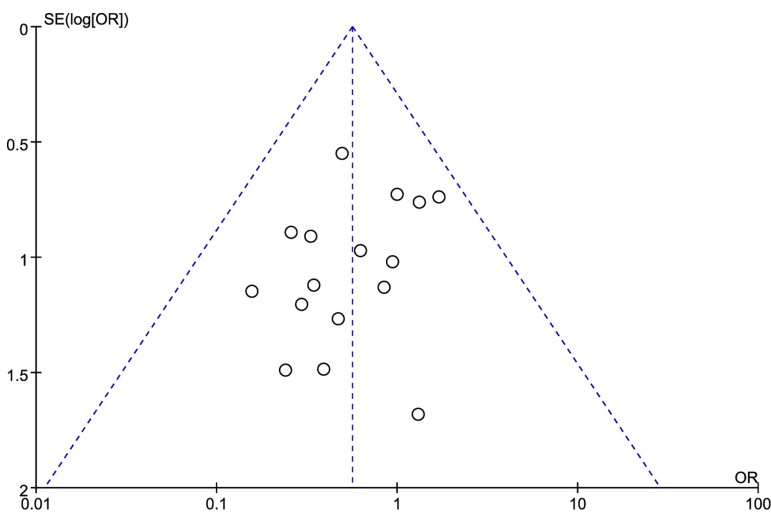

**Figure 3** **The relationship between lymphocyte to monocyte ratio (LMR) and overall survival (OS) in HNSCC.** A funnel map of the studies included in the meta-analysis.

| STUDY ID | Year | Cutoff | Harzard Ration M-H,Fixed,95% CI | DFS HR(95% CI) | p-value | Weight(%) |
|---|---|---|---|---|---|---|
| Jiechao Yang | 2018 | 2.98 | | 0.570(0.390-0.830) | 0.003 | 22.05 |
| Qian Song | 2019 | 3.17 | | 0.892(0.810-0.982) | 0.02 | 15.35 |
| Meng Ding | 2021 | 3.4 | | 0.460 (0.308-0.686) | <0.001 | 14.07 |
| Kosei Kubota | 2022 | 5 | | 1.370 (0.720–2.610) | 0.33 | 15.81 |
| Xiang Wu | 2021 | 4 | | 0.590(0.092-1.205) | <0.001 | 9.24 |
| Eltohami | 2018 | 4.85 | | 1.339 (1.028-1.744) | 0.0303 | 12.63 |
| Paolo Boscolo-Rizzo | 2022 | 3.76 | | 1.35(1.10-1.68) | <0.05 | 10.83 |
| Overall ($I^2$= 56.8%, P=0.020) | | | | 0.942 (0.631-1.382) | | 100 |

**Figure 4** **A forest map showing the relationship between LMR and DFS of patients with SCCHN in the 6 identified studies.**

most common types of cancer, of which 90% of cases are squamous cell carcinoma and mainly occur in the tongue, cheeks, gingiva, soft and hard jaws, oropharynx, and floor of the mouth (*Ghosh, Shah & Johnson, 2022*; *Mulder et al., 2021*). Despite improvements in surgery, radiotherapy, and chemotherapy treatments, the 5-year survival rate of patients with HNSCC has remained at 50% over the past decade. Around half of all patients develop distant metastasis after treatment, and there is a need for improved treatments in patients with advanced metastatic disease (*Chen et al., 2022*; *Cohen et al., 2019*).

**Table 4 A summary of the LMR endpoint statistics (DFS).**

| Study ID | Year | Cutoff | DFS HR [95% CI] | Weight (%) | P-value |
|---|---|---|---|---|---|
| Jiechao Yang | 2018 | 2.98 | 0.57 [0.39–0.83] | 22.05 | 0.003 |
| Qian Song | 2019 | 3.17 | 0.892 [0.810–0.982] | 15.35 | 0.02 |
| Meng Ding | 2021 | 3.4 | 0.460 [0.308–0.686] | 14.07 | <0.001 |
| Kosei Kubota | 2022 | 5 | 1.37 [0.72–2.61] | 15.81 | 0.33 |
| Xiang Wu | 2021 | 4 | 0.59 [0.092–1.205] | 9.24 | <0.001 |
| Eltohami | 2018 | 4.85 | 1.339 [1.028–1.744] | 12.63 | 0.0303 |
| Paolo Boscolo-Rizzo | 2022 | 3.76 | 1.35 [1.10–1.68] | 10.83 | <0.05 |
| Overall ($I^2$ = 56.8%, $P$ = 0.020) | | | 0.942 [0.631–1.382] | 100 | |

**Note:**

HR, hazard ratio; CI, confidence interval; OS, overall survival; DFS, disease-free survival; CSS, cancer-specific survival; PFS, progression-free survival.

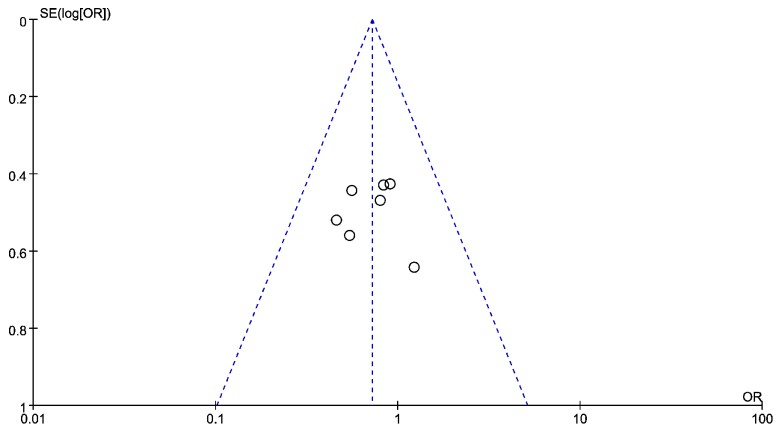

**Figure 5 The relationship between lymphocyte to monocyte ratio (LMR) and disease free survival (DFS) in HNSCC.** A funnel map of the studies included in the meta-analysis.

Studies have demonstrated the critical role of the immune system during tumor progression. The immune response and immune cell infiltration into the tumor microenvironment (TME) are closely related to tumor progression and patient prognosis (*Xia et al., 2021*). Tumor progression can be inhibited, and prognosis improved by reducing the role of host inflammatory cell mediators and immune regulatory pathways (*Zhu et al., 2022*). However, there is currently a lack of robust prognostic biomarkers for prognosis in patients with HNSCC. Tumor biomarkers are molecular signals that reflect the different biological processes and can have value in determining patient prognosis (*Fan et al., 2022*). Biomarkers should be specific, selective, and easily measured within a clinical environment (*Slagter et al., 2022*). Recently, prognostic biomarkers have been demonstrated for solid tumors, including inflammatory factors resulting from chronic immune stimulation, infections, and inflammatory sites (*Fairfield et al., 2021*). Tumor-related inflammation could accelerate tumor proliferation and metastasis by inhibiting apoptosis, inducing DNA damage and angiogenesis, and regulating anti-tumor

| STUDY ID | Year | Cutoff | Harzard Ration M-H,Fixed,95% CI | PFS HR(95% CI) | Weight(%) | *p-value* |
|---|---|---|---|---|---|---|
| JUN AOYAMA | 2021 | 1.994 | | 0.419(0.268-0.653) | 18.1 | <0.001 |
| Chuang | 2020 | 2.99 | | 2.127 (1.214–3.725) | 28.1 | 0.008 |
| Youfang Xun | 2019 | 0.18 | | 0.10 (0.04-0.24) | 24.71 | <0.001 |
| Michael Pogorzelski | 2021 | 3.7 | | 3.5 (2.8-4.2) | 29.79 | <0.001 |
| Overall (I$^2$= 64.8%, P=0.022) | | | | 0.932 (0.527-1.589) | 100 | |

**Figure 6** A forest map showing the relationship between the lymphocyte to monocyte ratio (LMR) and progression free survival (PFS) in patients with SCCHN in the three identified studies.

**Table 5** A summary of the LMR endpoint statistics (progression-free survival, PFS).

| Study ID | Year | Cutoff | PFS HR [95% CI] | Weight (%) | *P*-value |
|---|---|---|---|---|---|
| Jun Aoyama | 2021 | 1.994 | 0.419 [0.268–0.653] | 17.94 | <0.001 |
| Chuang | 2020 | 2.99 | 2.127 [1.214–3.725] | 27.87 | 0.008 |
| Youfang Xun | 2019 | 0.18 | 0.10 [0.04–0.24] | 24.5 | <0.001 |
| Michael Pogorzelski | 2021 | 3.7 | 3.5 [2.8–4.2] | 29.67 | <0.001 |
| Overall (I$^2$ = 64.8%, P = 0.022) | | | 0.932 [0.527–1.589] | 100 | |

**Note:**
HR, hazard ratio; CI, confidence interval; OS, overall survival; DFS, disease-free survival; CSS, cancer-specific survival; PFS, progression-free survival.

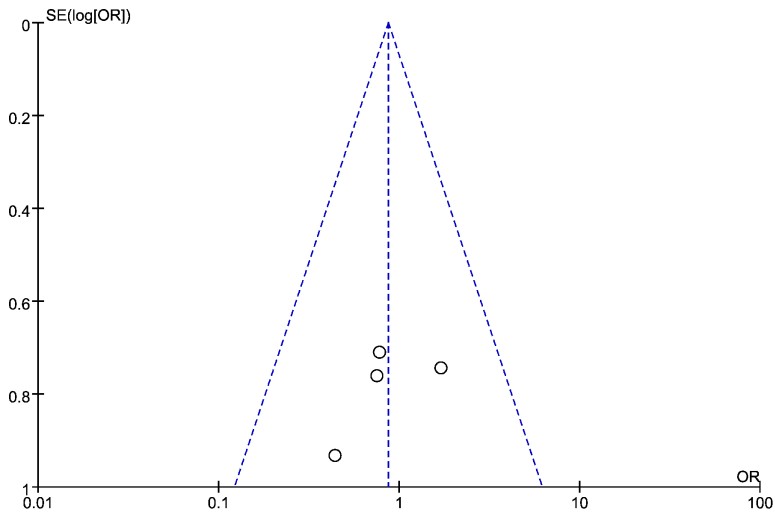

**Figure 7** The relationship between lymphocyte to monocyte ratio (LMR) and progression-free survival (PFS) in HNSCC. A funnel map of the studies included in the meta-analysis.

**Table 6 Summary of the meta-regression analysis of patient survival in HNSCC.**

| Variables | Number of documents | Number of cases | Merge HR [95% CI] | $P$ | $I^2$ (%) |
|---|---|---|---|---|---|
| Cut-off value | | | | | |
| <4 | 8 | 659 | 1.29 [1.05–1.59] | 0.401 | 58.7% |
| ≥4 | 6 | 677 | 1.34 [1.05–1.74] | 0.005 | 60.2% |
| Treatment method | | | | | |
| Comprehensive treatment | 7 | 1,066 | 1.45 [1.27–1.78] | 0.102 | 43.5% |
| Operation | 7 | 1,103 | 1.67 [1.28–2.09] | 0.090 | 0 |

Note:
HR, hazard ratio; CI, confidence interval; OS, overall survival; DFS, disease-free survival; CSS, cancer-specific survival; PFS, progression-free survival.

**Table 7 Subgroup analysis of the relationship between LMR and survival in patients with HNSCC.**

| Variables | Number of documents | Regression coefficient | Z value | HR [95% CI] | $P$ |
|---|---|---|---|---|---|
| Cut-off value | | | | | |
| <4 | 8 | 0.39 | 6.12 | 1.34 [1.05–1.74] | 0.005 |
| ≥4 | 6 | 0.35 | 5.89 | 1.76 [1.45–2.23] | 0.000 |
| Treatment method | | | | | |
| Comprehensive treatment | 7 | 0.44 | 8.21 | 1.45 [1.2–1.78] | 0.002 |
| Operation | 7 | 0.56 | 7.09 | 1.75 [1.21–1.73] | 0.009 |
| Sample size | | | | | |
| <200 | 9 | 0.05 | 0.57 | 1.67 [1.28–2.09] | 0.570 |
| ≥200 | 5 | 0.03 | 0.82 | 1.52 [1.30–1.98] | 0.993 |

Note:
HR, hazard ratio; CI, confidence interval; OS, overall survival; DFS, disease-free survival; CSS, cancer-specific survival; PFS, progression-free survival.

immunity (*D'Orso, Hyder & Mccann, 2020*). Lymphocytes and monocytes are well-established inflammatory biomarkers, and studies have shown that the LMR may have prognostic value in some tumor types (*Duan et al., 2022*). The LMR can be obtained from routine hematological examinations before treatment and is a simple, cost-effective, and reliable prognostic biomarker in some patients (*Caziuc et al., 2020*). The connection between the LMR and the prognosis of patients with HNSCC remains to be fully understood. This study performed a meta-analysis to assess the prognostic value of LMR in HNSCC patients.

After screening the articles, 16 studies were included in our analysis. We found that patients with an elevated LMR ratio had an improved prognosis. An increased LMR was correlated with improved OS (HR = 1.36% CI [1.14–1.62] $P$ = 0.018), DFS (HR = 0.942, 95% CI [0.631–1.382], $P$ = 0.02), and PFS (HR = 0.932, 95% CI [0.527–1.589], $P$ < 0.022). Since relatively few studies have investigated the LMR and CSS in patients with HNSCC, more prospective studies are needed to validate these findings.

Subgroup analysis showed that in HNSCC patients with an LMR of ≥4, the LMR had prognostic value. The data suggest that a low LMR level may be an independent risk factor for poor prognosis in HNSCC. Potentially, these data may be due to the critical function of these cell types in the anti-tumor immune response (*Ivankova et al., 2021*). Previous studies have indicated that tumor-infiltrating lymphocytes are solid prognostic indicators

in several cancers (*Xu et al., 2021*). The infiltration of CD4+ and CD8+T cells is a crucial part of the anti-tumor immune response that induces tumor cell apoptosis (*Sznurkowski et al., 2020*). The tumor inflammatory response can trigger immunosuppression and tumor cells can evade host immune surveillance (*Sumbayev et al., 2020*). Low lymphocyte counts have been observed in several human tumors and correlated with adverse clinical outcomes. Monocytes are also involved in tumorigenesis (*Radziszewski, 2021*), with evidence suggesting that tumor-associated macrophages (TAM) from monocytes are present at high density in tumor tissues. Macrophages promote tumor angiogenesis and an anti-immune response by emitting TNF-$\alpha$, vascular endothelial growth factor, and epidermal growth factor, eventually leading to tumor progression (*Cassetta & Pollard, 2020*). A previous meta-analysis was conducted to assess the prognostic effect of LMR in nasopharyngeal carcinoma (*Gao, Peng & Hu, 2022*). *Tham et al. (2018)* conducted a systematic review and meta-analysis to explore the correlation between LMR and prognosis in head and neck cancer. However, the study lacked high-quality prospective research. The function of the LMR in HNSCC remains to be fully understood. This study explains the observed outcomes demonstrating the prognostic value of lymphocytes and monocytes in HNSCC.

In summary, we found that a low LMR was correlated with poor prognosis in patients with HNSCC. Despite these exciting findings, our study had several limitations. Only studies published in English databases were searched, so data from other sources were not included. Also, we observed heterogeneity amongst the reported findings, which may be affected by age, sex, sample size, tumor stage, treatment, follow-up times, and other factors. Different studies had different cutoff values on the level of LMR, and different methods to measure the LMR may impact the results of our study. Further validation of our findings is required in extensive prospective studies to assess the value of LMR as a prognostic biomarker in patients with HNSCC.

## CONCLUSIONS

The meta-analysis indicates that lymphocytes and monocytes may have prognostic value in patients with HNSCC. A low LMR was associated with poor prognosis in patients with HNSCC. Hematological biomarkers are accessible, cost effective and could have potential in HNSCC. However, larger prospective, multicenter studies are required to validate our findings.

### Funding
The authors received no funding for this work.

### Competing Interests
The authors declare that they have no competing interests.

## Author Contributions

- Deyou Wei conceived and designed the experiments, analyzed the data, prepared figures and/or tables, and approved the final draft.
- Jiajia Liu conceived and designed the experiments, performed the experiments, authored or reviewed drafts of the article, and approved the final draft.
- Jipeng Ma performed the experiments, analyzed the data, prepared figures and/or tables, authored or reviewed drafts of the article, and approved the final draft.

## Human Ethics

The following information was supplied relating to ethical approvals (*i.e.*, approving body and any reference numbers):

The study was approved by the Yantai Hospital of Traditional Chinese Medicine, and was conducted in accordance to the tenets of the Declaration of Helsinki.

## Data Availability

The raw data is available in the Supplemental File.

## Supplemental Information

Supplemental information for this article can be found online at http://dx.doi.org/10.7717/peerj.16014#supplemental-information.

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
