# Peer review of "The value of lymphocyte to monocyte ratio in the prognosis of head and neck squamous cell carcinoma: a meta-analysis"

_PeerJ, doi:10.7717/peerj.16014_

## Round 0.1 · original submission · Minor Revisions

Both reviewers have recommended minor revisions and I agee with them. Please make the necessary modifications and re-submit your manuscript again.

Reviewer 1 ·

Basic reporting

The manuscript is short and concise and leaves little room for speculation, the results of the 14 studies analyzed in detail are heterogeneous and not overly conclusive, and this meta-analysis confirms the difficulty with this clinical parameter concerning its use for prediction or therapy selection.

Experimental design

no comment

Validity of the findings

no comment

Additional comments

1. In short, I suggest including studies from all over the world, not just Asia, provided these are available. If not, the authors must very clearly state why there is a location bias or population bias, and which consequences this may have for the scientific relevance of the manuscript.
2. Full name for the abbreviation might be offered at the first presence.
3. Results section with Begg’s funnel map and Figure 5 do not provide a detailed account of the results. This section needs to be detailed.
4. The section about meta-regression analysis needs to be substantially expanded upon instead of lists of results hidden in the supplement. What are the novel findings?
5. It is difficult to understand what exactly is meant by "yet the mechanism underpinning these observations remains unclear." in this study.
6. The format of Tables 4, 6, and 7 is different from the other tables. It is recommended that the unified table format be a three-line table.
7. Figure 2, Figure 4 and Figure 6 need to be recreated, it is suggested that the whole Figure should be a whole, rather than an adaptation of the Table.

Annotated reviews are not available for download in order to protect the identity of reviewers who chose to remain anonymous.

Reviewer 2 ·

Basic reporting

English language should be edited by a professional.

Experimental design

The model of meta-analysis should be confirmed, such as fixed or random?

Validity of the findings

The bias and limitations need to be disscussed in the disscussion part.

Additional comments

1. Head and neck squamous cell carcinoma (HNSCC) include oral, oropharynx, oropharyngeal larynx, and hypopharynx. The specific search keyword only “oral” and “head and neck”.
2. It is suggested to explore the source of heterogeneity with the results of subgroup analysis as supplement.
3. It will be better to show kappa for the selection and data extraction. Please show the data of kappa of agreement during the systematic searches.
4. Line 71, LMR is related to the prognosis of solid tumors such as tongue carcinoma. Is there any innovation in this article compared to the previous article?
5. Both lines 156 and 165 have A raised.
6. The authors should provide a clear rational for each analysis as there is a lack of integration.
7. There are multiple published correlations between LMR Ratio and head and neck cancer. I would suggest framing your findings with relevant discussions in context of those relationships.

---

## Round 0.2 · accepted · Accept

In carefully evaluating the content of this revised paper, I was satisfied with the responses and revisions made by the authors. The Reviewer's concerns have been well addressed. With the necessary revisions and improvements, the quality of this paper has been significantly improved. I believe that this revised manuscript is ready to be considered for publication in this journal.